# Attention-based Interpretable Deep Learning with Radiomic Features for Pulmonary Nodule Classification

**Doohyun Park**[1,*] ID                                    DOOHYUN.PARK@VUNO.CO
**Nahyuk Lee**[1]                                           NAHYUK.LEE@VUNO.CO
**Sungjoo Lim**[1]                                          SUNGJOO.LIM@VUNO.CO
[1] *Lung Vision AI Team, R&D Center, VUNO Inc., Republic of Korea*
[*] *Corresponding Author*

## Abstract

Pulmonary nodule classification is critical for early lung cancer screening, enabling timely intervention and evidence-based clinical decision-making. In this study, we introduce an attention-based interpretable deep learning framework that leverages mathematically predefined, handcrafted features derived from CT imaging for pulmonary nodule classification. In contrast to conventional convolutional neural networks (CNNs) that learn complex and often opaque feature representations, our approach prioritizes transparency and reproducibility by using statistically defined intensity features. The architecture is a lightweight multilayer perceptron (MLP) with channel-wise attention. The model was trained on an in-house dataset and validated on two publicly available external datasets: LUNA (n=1,122) and ISBI (n=220), achieving an area under the receiver operating characteristic curve (AUC) of 0.964 (95% confidence interval [CI]: 0.942–0.983) and 0.974 (95% CI: 0.964–0.984), respectively. The integration of channel-wise attention within the MLP architecture enables the model to explicitly learn and assign relative importance to each input feature, supporting feature-level interpretability.

**Keywords:** Attention, Feature Engineering, Deep Learning, Pulmonary Nodule.

## 1. Introduction

Lung cancer is one of the leading causes of cancer-related death worldwide. Despite recent advances, the five-year survival rate remains low at just 10–15% (Siegel et al., 2019). Early detection through computed tomography (CT) screening is critical for timely intervention and effective clinical management. In particular, the distinction between solid and ground-glass nodules (GGNs) is emphasized in recent global guidelines (Christensen et al., 2024).

Although convolutional neural networks (CNNs) have shown remarkable performance in medical image analysis by learning hierarchical feature representations directly from pixel data, they often suffer from limited interpretability due to the complexity of their latent feature space (Zhou et al., 2016; Lee et al., 2021). In contrast, radiomics provides a practical alternative by extracting predefined features from voxel-level intensity statistics within lesion masks, enabling more transparent and quantifiable representations (Aerts et al., 2014; Gillies et al., 2016; Park et al., 2022). Building on this, we investigate a lightweight deep learning model based on a multilayer perceptron (MLP) with channel-wise attention, using radiomic features as input to support interpretable and efficient classification of pulmonary nodules.

## 2. Materials and Methods

### 2.1. Dataset

This retrospective study used de-identified CT data curated by Segmed (`www.segmed.ai`). In this study, 1,416 nodules (1,277 solid, 139 GGN) were used for training and 472 (426 solid, 46 GGN) for internal validation. External validation was performed using two publicly available datasets: LUNA (n=1,122; 957 solid, 165 GGN) (Armato III et al., 2011; Setio et al., 2017) and ISBI (n=220; 151 solid, 69 GGN) (Balagurunathan et al., 2021).

### 2.2. Image Processing and Feature Extraction

Nodule masks were obtained using a commercial AI tool (VUNO Med®-LungCT AI™). CT volumes were resampled to $0.67{\times}0.67{\times}1.0$ mm³, from which 22 first-order and volumetric features were extracted from the axial, coronal, and sagittal slices with the largest cross-sectional area per nodule, yielding 66 handcrafted features.

### 2.3. Model Architecture

As shown in Figure 1, we implemented a lightweight MLP with two hidden layers of 128 units, augmented with channel attention inspired by the Squeeze-and-Excitation (SE) block (Hu et al., 2018) in Convolutional Block Attention Module (CBAM) (Woo et al., 2018). The model receives 66 z-score normalized radiomic features as input.

To enhance interpretability, we used a channel-wise attention mechanism that adaptively reweights input features. Average and max pooling across the batch dimension capture global context, which is passed through a shared two-layer fully connected network with ReLU activation and a sigmoid gate. The resulting attention vector scales the input features via element-wise multiplication, emphasizing informative channels. These attention-refined features are passed through two ReLU-activated hidden layers, followed by a sigmoid output layer for binary classification. This architecture enables direct assessment of feature importance through learned attention weights while maintaining a compact structure suitable for deployment in resource-limited settings.

## 3. Results

The proposed model demonstrated robust performance across all datasets. On the internal validation set, the model achieved an area under the receiver operating characteristic curve (AUC) of 0.972 (95% CI: 0.953–0.987). On the LUNA external dataset, the model achieved an AUC of 0.964 (95% CI: 0.942–0.983), and 0.974 (95% CI: 0.964–0.984) on the ISBI dataset. Table 1 summarizes the model's classification performance.

Table 1: Performance summary across datasets.

| Dataset | AUC | Accuracy | Sensitivity | Specificity | F1-score |
|---|---|---|---|---|---|
| Internal | 0.972 | 0.947 | 0.960 | 0.826 | 0.970 |
| External (LUNA) | 0.974 | 0.949 | 0.975 | 0.800 | 0.970 |
| External (ISBI) | 0.964 | 0.909 | 0.927 | 0.870 | 0.933 |

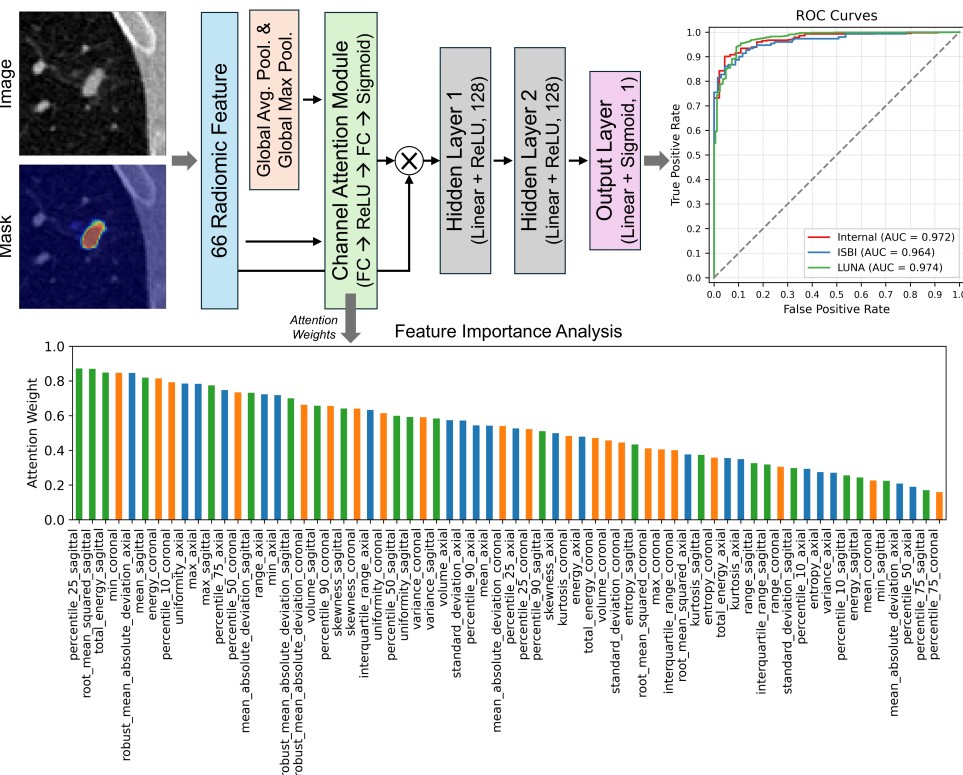

Figure 1: Architecture of the proposed model, receiver operating characteristic curves for validation sets, and feature importance analysis.

To interpret the model, we analyzed the learned channel-wise attention weights. Interestingly, unlike our previous study using machine learning model (Park et al., 2025), where the features of the axial view were found to be more informative, the MLP assigned a relatively higher importance to the features extracted from the sagittal view. This suggests that deep learning and classical machine learning models may capture distinct representational patterns even when trained on the same set of handcrafted features. Despite this difference, both models demonstrated high classification performance, reinforcing the robustness of radiomic features.

## 4. Discussion

This study demonstrates that interpretable and robust deep learning models can be achieved even with lightweight architectures by leveraging well-defined radiomic features in place of trainable image encoders. The proposed MLP with channel-wise attention uses predefined, mathematically grounded features derived from CT scans, achieving strong predictive performance without reliance on complex and computationally intensive deep learning models. The attention mechanism enables direct interpretation of feature importance, illustrating the value of combining domain-informed feature engineering with model-level transparency.

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
