# OpenReview forum: "Attention-based Interpretable Deep Learning with Radiomic Features for Pulmonary Nodule Classification"
_MIDL.io/2025/Short_Papers — MIDL 2025 - Short Papers_

### Official Review · Reviewer_Xye3 · 2025-04-25

**Rating:** 3
**Confidence:** 3

**Summary:**

This paper proposes an attention-based method for pulmonary nodule classification, which utilizes predefined radiomic features extracted from CT images. A lightweight multi-layer perceptron combined with a channel attention mechanism is employed for classification. The model is trained on an internal dataset and evaluated on two external datasets, achieving high AUC scores across all evaluations.

**Strengths:**

The proposed method has low computational requirements, as it employs a lightweight MLP architecture, making it suitable for deployment in resource-constrained environments.

The model offers a certain degree of interpretability. By incorporating a channel attention mechanism, it can assign importance weights to each input feature, enhancing its explainability.

The experimental datasets are diverse, including one internal dataset and two publicly available external datasets, which helps demonstrate the generalizability of the model.

**Weaknesses:**

Presentation issue: Figure 1 is not referenced in the main text, and the correspondence between the figure and the narrative is unclear, which affects the overall clarity and completeness of the presentation.

Lack of comparative experiments: The paper does not include direct comparisons with recent state-of-the-art methods for pulmonary nodule classification, making it difficult to fully assess the competitiveness of the proposed approach.

Insufficient training details: Key training settings, such as the number of training epochs, learning rate, and computational environment, are not provided. This lack of detail hinders the reproducibility of the model.

Model limitations: Although the lightweight MLP architecture offers computational efficiency, its relatively simple structure may limit its ability to effectively handle morphologically complex pulmonary nodules. This raises concerns about the model’s performance in more challenging real-world scenarios.

---

### Decision · Program_Chairs · 2025-05-01

Accept